# Ecology of *Endozoicomonadaceae* in three coral genera across the Pacific Ocean

Corentin Hochart [1], Lucas Paoli [2], Hans-Joachim Ruscheweyh [2], Guillem Salazar [2], Emilie Boissin [3], Sarah Romac[4], Julie Poulain[5,6], Guillaume Bourdin [7], Guillaume Iwankow[3], Clémentine Moulin[8], Maren Ziegler[9], Barbara Porro[10,11], Eric J. Armstrong [3], Benjamin C. C. Hume[12], Jean-Marc Aury [5,6], Claudia Pogoreutz [3,12], David A. Paz-García[13], Maggy M. Nugues [3], Sylvain Agostini [14], Bernard Banaigs[3], Emmanuel Boss [15], Chris Bowler [6,16], Colomban de Vargas[4,6], Eric Douville [17], Michel Flores [18], Didier Forcioli[10,11], Paola Furla [10,11], Eric Gilson [10,11,19], Fabien Lombard [6,20,21], Stéphane Pesant[22], Stéphanie Reynaud [11,23], Olivier P. Thomas[24], Romain Troublé [8], Patrick Wincker [5,6], Didier Zoccola [11,23], Denis Allemand [11,23], Serge Planes[3,6], Rebecca Vega Thurber [25], Christian R. Voolstra [12], Shinichi Sunagawa [2] & Pierre E. Galand [1,6] ✉

Health and resilience of the coral holobiont depend on diverse bacterial communities often dominated by key marine symbionts of the *Endozoicomonadaceae* family. The factors controlling their distribution and their functional diversity remain, however, poorly known. Here, we study the ecology of *Endozoicomonadaceae* at an ocean basin-scale by sampling specimens from three coral genera (*Pocillopora*, *Porites*, *Millepora*) on 99 reefs from 32 islands across the Pacific Ocean. The analysis of 2447 metabarcoding and 270 metagenomic samples reveals that each coral genus harbored a distinct new species of *Endozoicomonadaceae*. These species are composed of nine lineages that have distinct biogeographic patterns. The most common one, found in *Pocillopora*, appears to be a globally distributed symbiont with distinct metabolic capabilities, including the synthesis of amino acids and vitamins not produced by the host. The other lineages are structured partly by the host genetic lineage in *Pocillopora* and mainly by the geographic location in *Porites*. *Millepora* is more rarely associated to *Endozoicomonadaceae*. Our results show that different coral genera exhibit distinct strategies of host-*Endozoicomonadaceae* associations that are defined at the bacteria lineage level.

Tropical coral reefs are hotspots of biodiversity in the world's oceans. They provide food and shelter for many marine animals, and are important to humans who benefit from the reefs' goods and services[1]. In recent decades, global warming and human activities have, however, increasingly threatened coral reefs[2–5]. Their resilience and adaptability rely heavily on their ecosystem engineers, the corals, and their ability to resist damaging perturbations. In turn, the coral's adaptive capabilities are anchored in its relationship with associated microorganisms that together with the host form the holobiont[6–9].

Coral holobionts include complex microbial communities composed of viruses, fungi, archaea, bacteria and micro-algae[10]. The symbiosis between the Symbiodiniaceae algae and corals is the best characterized association within the coral holobiont[11], but bacteria are also important partners. Coral-associated bacteria contribute significantly to the health of the host by participating in nutrient acquisition, metabolic (re)cycling, and protection against pathogens[6,7,12,13]. Understanding the composition, the diversity and the functions of coral-associated microorganisms can thus provide clues about the health status, but also the resilience and adaptive capabilities of corals[14]. Among the large diversity of coral-associated bacteria[15] some are noteworthy because of their abundance and commonness. Bacteria from the *Endozoicomonadaceae* family (*Gammaproteobacteria, Oceanospirillales*) are frequent in corals[7,16], but also in a number of other marine invertebrates such as sponges[17,18], bivalves[19], ascidians[20] or gorgonians[21]. Their global distribution in a large number of hosts make them potential key bacterial symbionts in the marine ecosystem[16]. An strict host-bacteria interdependency remains, however, to be demonstrated since *Endozoicomonadaceae* have some characteristics of non-obligate symbionts including possible free-living stages[22], and relatively large genome sizes[23,24].

The family *Endozoicomonadaceae* was described only recently[25] and comprises the genera *Parendozoicomonas*[25], *Kistimonas*[26], *Sansalvadorimonas*[27] and *Endozoicomonas*[28]. Although the definition of the genus *Endozoicomonas* is recent, retrospective analysis show that it was commonly detected in early coral microbiome studies[29–32], and it continues to be frequently reported in a number of different coral species[23,33–42]. Different corals may harbor different *Endozoicomonadaceae* 16S rRNA types that could show patterns of co-phylogeny with the coral host[41,43]. The patterns of host-symbiont associations have also been hypothesized to be linked to the reproductive mode of the host[43], or the environmental condition[44], while other studies suggest a correlation between *Endozoicomonas* and Symbiodiniaceae[45]. The question of host-specificity remains open, however, since some *Endozoicomonadaceae* types could be shared between different hosts[41], different *Endozoicomonadaceae* types can dominate within a same host[43], and changes in their relative abundance seems to be independent of the symbiotic algae[46]. If the host is not the driver of microbiome composition, external factors could play a role[47]. *Endozoicomonadaceae* communities could be shaped by environmental factors as they were shown to be less abundant in *Acropora millepora* at lower seawater pH[48,49], or during increased temperatures and subsequent bleaching[31,50], as well as anthropogenic impact and habitat suitability[44,51]. In *Porites astreoides*, lesioned colonies also contained fewer *Endozoicomonas* sequences, compared to non-lesioned colonies[52]. These findings suggest that *Endozoicomonas* are underrepresented in stressed corals[14], however, it's not always the case. In *Pocillopora verrucosa* and *Acropora hemprichii*, although *Endozoicomonadaceae* decreased at sites impacted by sedimentation, they increased at sites impacted by municipal wastewater[39]. *Endozoicomonadaceae* also increased in abundance in *Porites* spp. under lower pH[53], and under natural stressful conditions of shallow hydrothermal vent[54], and were not impacted by coral bleaching or severe tissue sloughing in *P. verrucosa* in response to eutrophication[55]. *Endozoicomonadaceae* also proliferated after warm summer months in French Polynesia[42]. These contrasted observations between individual studies conducted locally illustrate the need for a large-scale approach on multiple coral species to unveil *Endozoicomonadaceae* diversity and biogeography, and better understand the factors controlling their presence and community composition. The widespread prevalence of *Endozoicomonadaceae* in corals indicates that they are important for the host, and the analyses of their genomes have given some clues regarding their potential interactions and the establishment of symbiosis with corals[56]. *Endozoicomonadaceae* have large genome sizes that suggest a possibility for a non-symbiotic living stage before host invasion[23,24,57].

Host infection could be promoted by type III secretion system, like the one detected in a strain of *Endozoicomonas montiporae*[57], and by potential effector proteins that that could help interactions[23,56,58]. A comparative analysis of several *Endozoicomonadaceae* genomes showed an enrichment of genes associated with carbon sugar transport and utilization, protein secretion, and synthesis of amino acids that could potentially be transferred to the host[43]. In addition, *Endozoicomonas acroporae* has the potential to degrade dimethylsulfoniopropionate (DMSP), which bacteria could use as a carbon source[24]. Various studies also suggested that *Endozoicomonadaceae* play a role in regulating the overall microbiome structure either by direct competition with other bacteria, or by producing antimicrobial compounds[48,59]. The extent of metabolic capabilities of *Endozoicomonadaceae* across species remains, however, to be mapped, and the variability of these pathways at ocean scale is not known.

In this work, we study the ecology of *Endozoicomonadaceae* at an ocean-basin scale and in different coral genera. Our hypothesis was that different corals may have evolved different strategies of host-bacteria relationships. To assess this, we conducted an unprecedented campaign to methodically sample three globally distributed coral species. Based on morphology, we targeted the complex *Pocillopora meandrina*, the robust *Porites lobata*, and the fire coral *Millepora platyphylla* on 99 reefs from 32 islands across the Pacific Ocean. We precisely described the diversity of *Endozoicomonadaceae* in more than 2400 coral colonies by metabarcoding the 16S rRNA gene. In addition, we sequenced 270 metagenomes to assemble high-quality draft genomes of novel *Endozoicomonadaceae* species, and putative lineages. Here we show that the most common *Endozoicomonadaceae* lineage, found in *Pocillopora*, is globally distributed and has distinct metabolic capabilities, including the synthesis of amino acids and vitamins not produced by the host. The other lineages are structured partly by the host genetic lineage in *Pocillopora* and mainly by the geographic location in *Porites*. Our results demonstrate that different coral genera exhibit distinct strategies of host-*Endozoicomonadaceae* associations that are defined at the bacteria lineage level.

## Results

### *Endozoicomonadaceae* distribution and abundance in corals and seawater across the Pacific Ocean

Overall, *Endozoicomonadaceae* ASVs were detected in 99% of the coral samples (n = 2447). Despite their prevalence, their relative abundance varied greatly between samples (Fig. 1b). *Pocillopora* had the highest proportion of ASVs affiliated to *Endozoicomonadaceae* (53% of all *Pocillopora* sequences) followed by *Porites* (30%) and then *Millepora* (11%). *Pocillopora* also had the highest *Endozoicomonadaceae* richness, followed by *Porites* and then *Millepora* (Fig. 1c). In *Pocillopora*, *Endozoicomonadaceae* were present in all islands, but with low relative abundances in Coïba (I02), Malpelo (I03), and Guam (I15). In *Porites*, *Endozoicomonadaceae* were not detected east of Rapa Nui (I04), and in Kiribati (I13) and Southwest Palau Islands (I25). In *Millepora*, *Endozoicomonadaceae* were almost absent in corals from islands located east of Fiji (I18), with the exception of Samoa (I10), and almost absent from Guam (I15) and Ogasawara Islands (I16). We observed high variations in relative abundance between sites in some islands, like in Las Perlas (I01), Cook (I08) and Kiribati (I13) for *Pocillopora*, and Gambier (I06), Upolu (I10) and Chuuk Island (I14) for *Porites*, and New Caledonia (I21), Southwest Palau Islands (I25) for *Millepora*. In other cases, there was a high relative abundance and low variability between sites (Fig. 1b). At any given site, the presence of *Endozoicomonadaceae* in one coral genus was not predictive of its presence in the other genera (e.g., Rapa Nui I04 or Moorea I07), although at other sites, *Endozoicomonadaceae* were present and abundant in all three coral genera (e.g., Chesterfield I20 and New Caledonia I21).

Overall, asv0000001 was the most abundant *Endozoicomonadaceae* (12% of the sequences in the dataset) and the most common

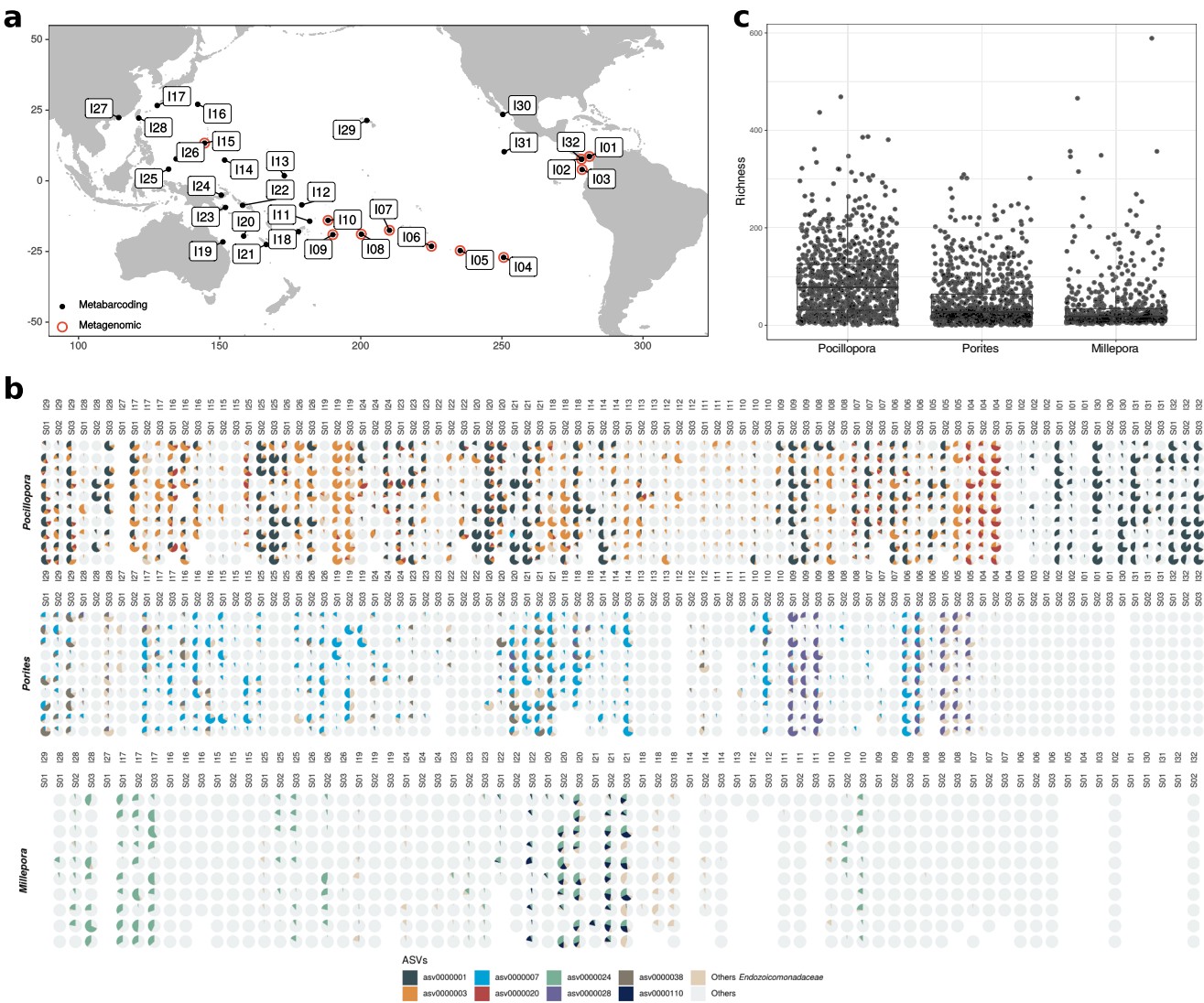

**Fig. 1 | *Endozoicomonadaceae* distribution and abundance in corals across the Pacific. a** Map showing the islands sampled during the *Tara* Pacific expedition. Black and red symbols indicate whether samples were analyzed by metabarcoding (16S rRNA gene) and/or metagenomics. **b** Proportion of the most abundant *Endozoicomonadaceae* ASVs in *Pocillopora, Porites*, and *Millepora* sampled across 99 reefs from 32 islands across the Pacific Ocean. The islands are ordered longitudinally from west to east. **c** *Endozoicomonadaceae* community richness in the corals *Pocillopora* (*n* = 959 samples), *Porites* (*n* = 944), and *Millepora* (*n* = 544). The box plot horizontal bars show the median value, the box indicates the first and third QRs, and the whiskers indicate 1.5*IQR. Source data are provided as a Source Data file.

(found in 78% of the samples) (Fig. 2). *Pocillopora* had 3 dominant ASVs (abundance >1%, asv0000001, asv0000003, asv0000020) of which asv0000020 was more abundant at Rapa Nui (I04) and asv0000003 at the Great Barrier Reef (I19) (Fig. 1). *Porites* also had 3 dominant ASVs (asv0000007, asv0000028, asv0000038); asv0000028 dominated at Ducie Island (I05) and Niue (I09). *Millepora* had two dominant ASVs (asv0000024, asv0000110), of which asv0000110 was found mostly in Chesterfield (I20), New Caledonia (I21) and Solomon Islands (I22) (Fig. 1). All dominant ASVs had a broad geographic distribution since the same ASVs were found across the entire Pacific Ocean. Each of the ASVs defined as dominant in one coral host were also detected at very low proportions (>0.1%) in the other coral species (Fig. 2).

When the relative abundance of *Endozoicomonadaceae* in corals was low, the diversity of the overall microbial community reached higher values (Supplementary Fig. 1a). In *Pocillopora*, corals with low abundance of *Endozoicomonadaceae* were often dominated by *Flavobacteriaceae* (Supplementary Fig. 1b). In *Porites*, there were more *Kiloniellaceae* and *Rhodobacteraceae*, while in *Millepora* there were more *Spirochaetaceae*.

We also looked for *Endozoicomonadaceae* ASVs in the water surrounding the *Pocillopora* colonies, in surface water over the reef, and in the surface water off the island (Fig. 3). *Endozoicomonadaceae* sequences were detected in the water, but their relative abundance decreased rapidly with increasing distance from the colonies (Fig. 3a). While *Endozoicomonadaceae* represented on average 30% of the bacteria in *Pocillopora*, their relative abundances decreased to ~0.5% in coral surrounding water, 0.05% in surface water over the reef, and 0.001% in surface water off the island (Fig. 3a). Additionally, we observed a significant positive correlation (*r* = 0.52) between the relative abundance of *Endozoicomonadaceae* in *Pocillopora* and in the colony surrounding water (Fig. 3b). The *Endozoicomonadaceae* ASVs that were abundant in *Pocillopora* were also detected in the water, but in varying proportions (Fig. 3c). When moving away from the *Pocillopora* colonies, the proportion of typical *Pocillopora Endozoicomonadaceae* (asv0000001, asv0000003 and asv0000020) decreased, while the proportion of other *Endozoicomonadaceae* increased. In some cases, *Endozoicomonadaceae* were not detected at all in the surface water off the island.

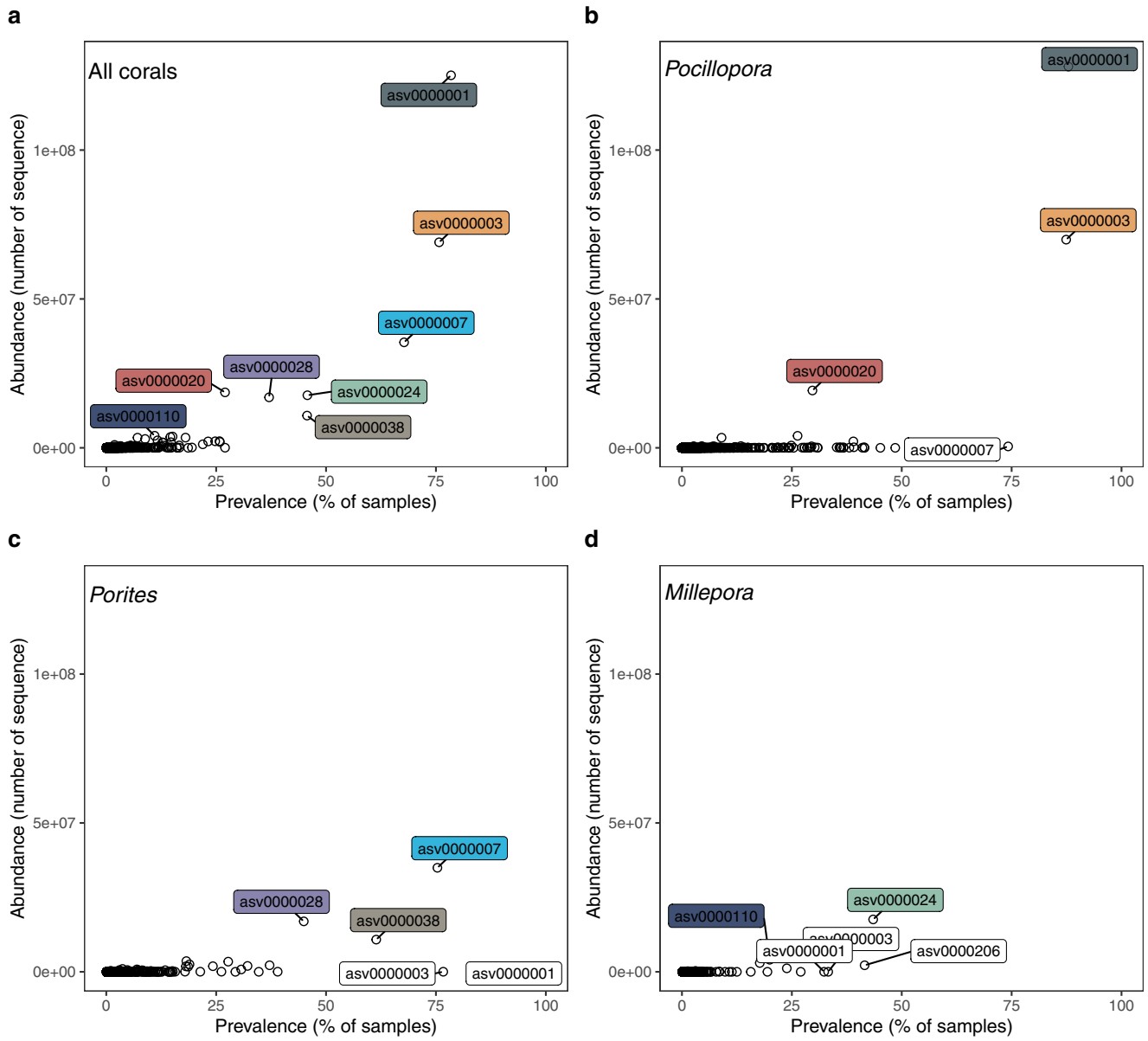

**Fig. 2 | Prevalence and relative abundance of Endozoicomonadaceae.** Endozoicomonadaceae ASVs found in **a** all corals, **b** Pocillopora, **c** Porites and **d** Millepora.

## Correlation between *Endozoicomonadaceae*, environment, islands, host genetic lineages, Symbiodiniaceae symbionts, and phylotypic biomarkers

There was no strong correlation between *Endozoicomonadaceae* relative abundance at the family level and chlorophyll a, pH, phosphate, silanol, sea surface salinity and temperature in *Pocillopora* and *Millepora* (Fig. 4a). In *Porites, Endozoicomonadaceae* had the highest correlation to salinity. At the ASV level, however, all correlation values were low, but there were differences within coral genera. In *Pocillopora*, asv0000001 had the strongest positive correlation to sea surface temperature and was negatively associated with pH, while asv0000003 and asv0000020 had the highest correlations to salinity and were negatively correlated to SiOH (Fig. 4a). In *Porites*, asv0000007 had the highest positive correlation to salinity, while asv0000028 and asv0000038 only had very low correlation values to environmental conditions. In *Millepora*, asv0000024 had the highest positive correlation to SiOH and asv0000110 only had low correlation values (Fig. 4a).

Information about assemblages of Symbiodiniaceae, phylotypic biomarkers indicative of coral health, and host genetic lineages was obtained from a subset of 11 islands (Fig. 1a−red circle). The host genetic lineages identified through genome-wide SNP calling (material and methods) separated *Pocillopora* in five genetic lineages (Poc-SVD1 to Poc-SVD5), *Porites* in nine lineages (Por-K1a to Por-K3d) and *Millepora* in six lineages (Mil-SVD1 to Mil-SVD6). Overall, the composition of *Endozoicomonadaceae* communities was best explained by the genetic lineage of the host for both *Pocillopora* and *Porites* (Fig. 4b). Symbiodiniaceae had low significance with regard to *Endozoicomonadaceae* assemblage in *Pocillopora*, and the environment had low significance in *Porites*. Phylotypic biomarkers were always the lowest explanatory factor.

At the ASV level, variance partitioning showed that individual ASVs had a higher proportion of variance explained than overall *Endozoicomonadaceae* (Fig. 4c). For *Pocillopora*, asv0000001 and asv0000003 had the largest proportion of their variation explained by the factor 'island', while asv0000020 was explained mostly by the genetic lineage of the host (Fig. 4c). In *Porites*, the overall *Endozoicomonadaceae* relative abundance was best explained by the host genetic lineage, but all individual ASVs (asv0000007, asv0000028 and asv0000038) had most of their variance explained by island.

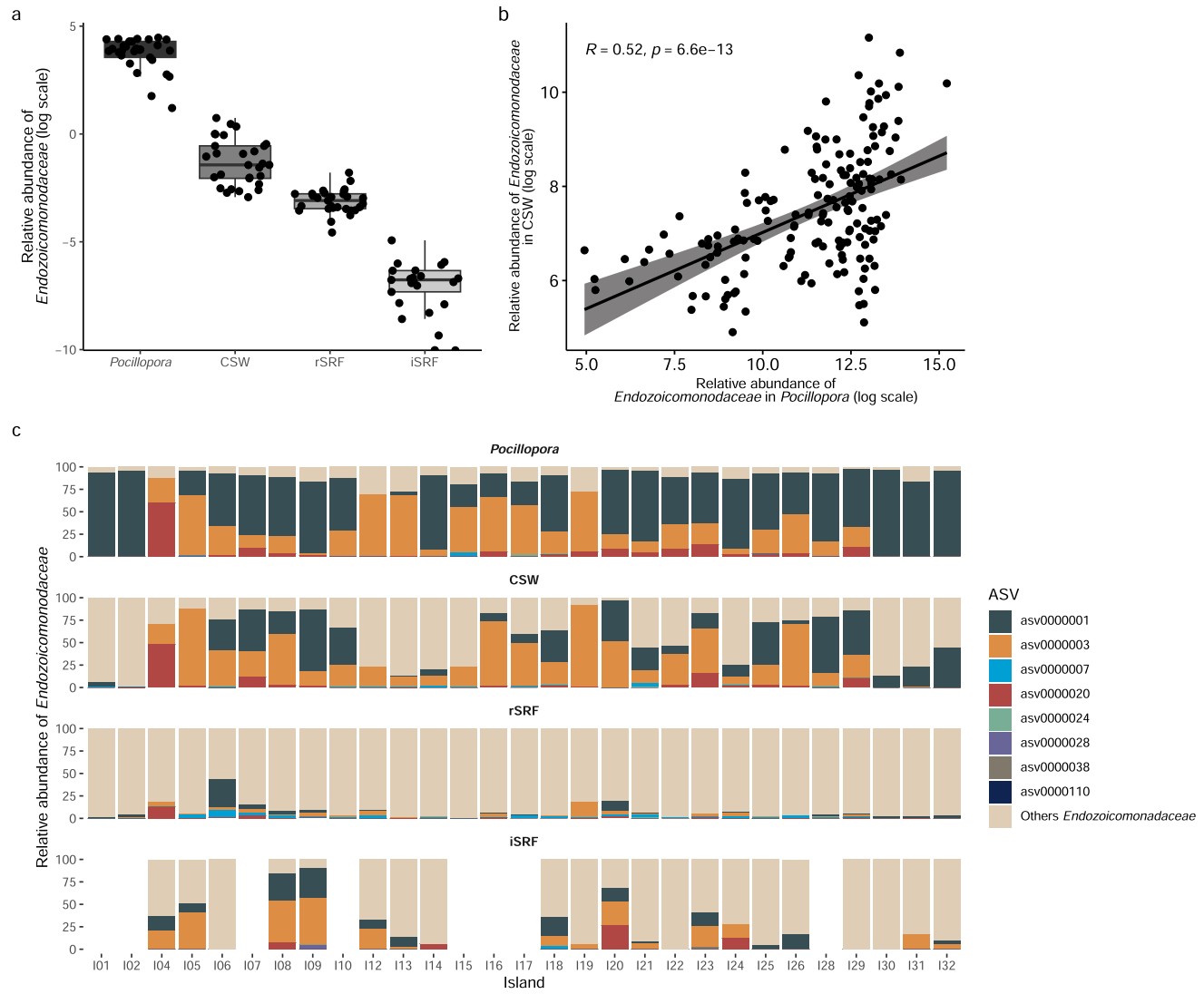

**Fig. 3 | Relative abundance of *Endozoicomonadaceae* in *Pocillopora* corals and the surrounding environment. a** *Endozoicomonadaceae* relative abundance in the coral *Pocillopora* host, the colony surrounding water (CSW), the surface water over the reef (rSRF), and the surface water off the island (iSRF) in the 0.2–3 μm planktonic size fraction. The box plot horizontal bars show the median value, the box indicates the first and third QRs, and the whiskers indicate 1.5*IQR. **b** Spearman correlation between relative abundance of *Endozoicomonadaceae* in *Pocillopora* and in colony surrounding water (CSW). The lines represent the result of linear regression with the gray shadow as the 95% confidence interval. A two-sided *p*-value is given. **c** Relative abundance of the dominant *Endozoicomonadaceae* ASVs in the different sampling environments at each island. Source data are provided as a Source Data file.

Asv0000038 had only a small portion of the variance explained by the factors that we tested. In *Millepora*, the Symbiodiniaceae assemblage was the strongest explanatory factor for both asv0000024 and asv0000110.

On the 11 islands we could also correlate individual ASV relative abundance to individual coral biomarkers. Different ASVs showed distinct correlation patterns (Supplementary Fig. 2). In *Pocillopora*, asv0000001 had strongest positive correlations to Symbiodiniaceae cell content and negatively to protein carbonylation (Supplementary Fig. 2). Asv000003 was positively correlated to total antioxidant content and asv0000020 to protein ubiquitination. In *Porites*, asv0000007 was most positively correlated to Symbiodiniaceae content, and asv0000038 to protein carbonylation (Supplementary Fig. 2).

## Genomic characteristics and phylogeny of *Endozoicomonadaceae*

A total of 270 coral metagenomes were sequenced from a subset of 11 islands covering 33 reefs (Fig. 1a). We reconstructed a total of 24 MAGs belonging to the *Endozoicomonadaceae* family. They had a completeness ranging from 51.2% to 93.3%, and contamination levels from 0% to 4.9% (Supplementary Data 1). To conduct comparative genomics analyses, all publicly available *Endozoicomonadaceae* genomes were downloaded from NCBI or recovered from the literature (*n* = 30). The taxonomic annotation of the MAGs and the phylogenetic tree based on 71 marker genes separated the *Endozoicomonadaceae* into five distinct genera: *Endozoicomonas*, *Parendozoicomonas*, *Kistimonas*, *Sansalvadorimonas*, and one unnamed genus (Fig. 5, Supplementary Data 1). Our *Tara* Pacific MAGs fell within two of these genera: the *Endozoicomonas* genus for *Millepora* and *Pocillopora* MAGs, and the *Parendozoicomonas* genus for *Porites* MAGs. All *Endozoicomonadaceae* genomes had similar coding density (80.8–91.6%), but distinct guanine-cytosine (GC) contents (33.2–57.3%) between genera (Supplementary Data 1). *Endozoicomonas* and *Parendozoicomonas* had similar GC contents with, respectively, 48.5% and 48.2%, while the only *Kistimonas* and *Sansalvadorimonas* genomes available had 57.3% and 50.3%, respectively (Supplementary Data 1).

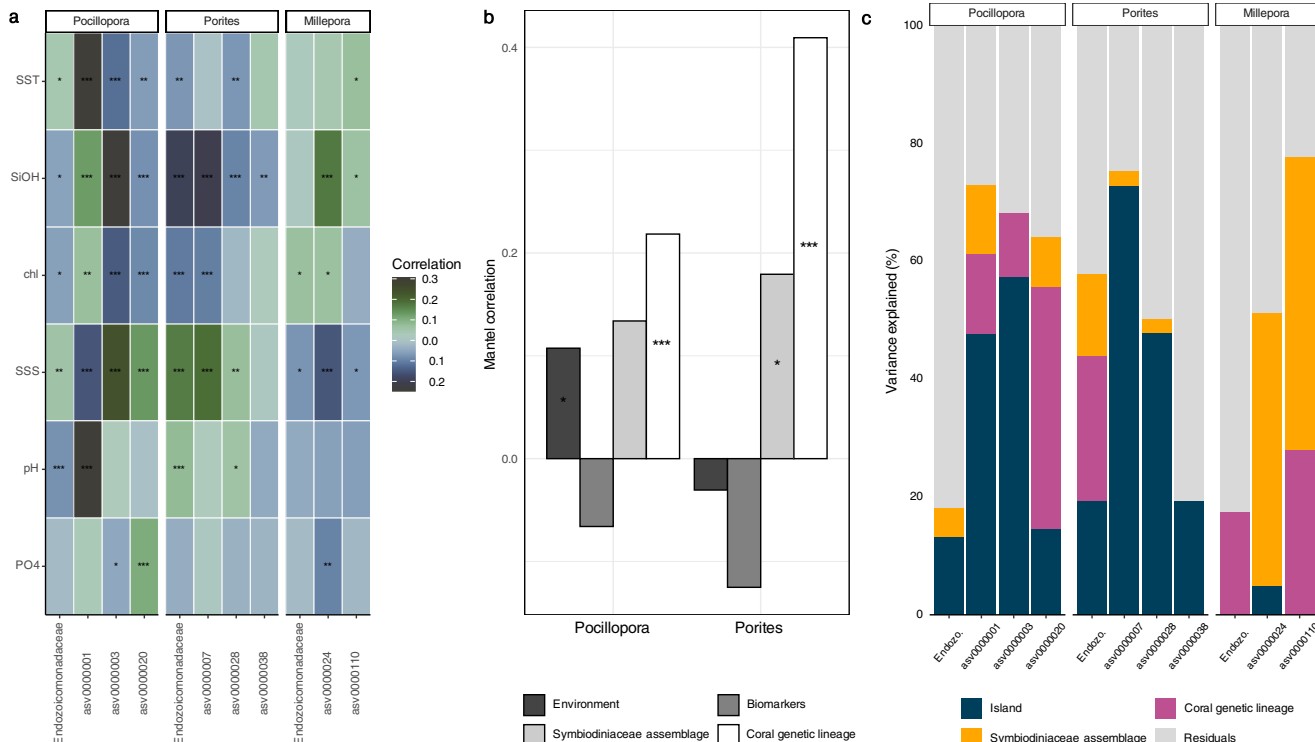

**Fig. 4 | Correlation between *Endozoicomonadaceae*, environment, islands, host genetic lineages, Symbiodiniaceae symbionts, and phylotypic biomarkers.**
**a** Kendall correlations across 32 islands between abundant *Endozoicomonadaceae* ASVs and environmental factors. *Pocillopora* ($n = 889$ samples), *Porites* ($n = 903$), and *Millepora* ($n = 515$). SST: Sea Surface Temperature; SiOH: Silanol; chl: Chlorophyll a; SSS: Sea Surface Salinity; pH; PO4: phosphate. **b** Mantel correlation across 11 islands between *Endozoicomonadacea* community composition and environmental conditions, phylotypic biomarkers, Symbiodiniaceae assemblages and coral host genetic lineages for the two coral genera *Pocillopora* ($n = 77$) and *Porites* ($n = 80$) phylotypic biomarkers data for *Millepora* are missing. **c** Variance partition analysis explaining the variance of the relative abundance of *Endozoicomonadaceae* and individual ASVs across 11 islands in relation to island, coral genetic lineages and Symbiodiniaceae assemblages in *Pocillopora* ($n = 96$), *Porites* ($n = 99$),and *Millepora* ($n = 55$). Source data are provided as a Source Data file.

The *Parendozoicomonas* genomes formed a monophyletic group with average ANI and AAI values of 73.80% (73.41–74.12%) and 71% (70.33–71.41%), respectively (Supplementary Fig. 3, Supplementary Data 2). The *Endozoicomonas* genomes also formed a well defined group, and they had average ANI and AAI values of 76% (71.08–99.99%) and 73% (60.96–99.99%), respectively. The genus *Sansalvadorimonas* has only one representative, *Sansalvadorimonas verongulae*. Because its genome is not present in the latest version of GTDB database, it was first annotated as *Parendozoicomonas*, but our genomic analysis confirmed its position as a distinct genus.

Our *Tara* Pacific MAGs from *Pocillopora*, *Porites*, and *Millepora* were divided into three well-supported monophyletic clusters that corresponded to the three coral hosts (Fig. 5a, Supplementary Fig. 3). *Endozoicomonas* MAGs obtained from *Pocillopora* all grouped with sequences from the *Pocilloporidae* family earlier obtained from one *P. verrucosa* and two *S. pistillata*[23]. The two *Endozoicomonas* genomes from *S. pistillata* represent two distinct species (93% ANI similarity). *Parendozoicomonas* MAGs from *Porites* clustered with a MAG obtained earlier from *Porites lutea*[12]. It was previously annotated as *Endozoicomonas*, but our analysis shows that it is a *Parendozoicomonas* with closer similarity to *Parendozoicomonas* from *P. haliclonae* (73.8% ANI and 70.69% AAI) than with other *Endozoicomonas* genomes (71.83% ANI and 60.50% AAI). *Millepora* MAGs were distantly related to all genomes published to date (Fig. 5a).

Although *Endozoicomonas* genomes from the same host species clustered together, there was no clear pattern of co-phylogeny between the bacteria and their host. *Endozoicomonas* genomes from different coral genera were often more related to genomes from other marine invertebrates (Supplementary Fig. 3). For instance, the *Endozoicomonas* symbionts of two *Acropora species*, *A. humilis* and *A.*

*muricata*, were separated in two distinct clades and were more closely related to a sponge and an ascidian symbiont than to each other. There were, however, exceptions and *Endozoicomonas* symbionts from *S. pistillata*, *P. verrucosa*, and *P. meandrina* clustered according to host phylogeny.

In the MAGs from this study, ANI was >95% in coral host-specific clusters, which corresponds to the threshold used for species delineation[60]. The specI method confirmed that each coral species harbored one specific *Endozoicomonadaceae* species (Supplementary Data 1). We could also delineate subclades within host-specific *Endozoicomonadaceae* species. Within *Pocillopora*, the MAGs were separated into three subclades that corresponded to three distinct *Pocillopora* genetic lineages (Poc-SVD5, Poc-SVD1 and Poc-SVD3) (Fig. 5c). For *Porites*, the MAGs separated into 6 subclades (Fig. 5c). The subclades that contained most MAGs corresponded exclusively to the *Porites* genetic lineages K3b. For *Millepora*, all MAGs shared a high identity and thus belonged to the same group.

Based on the phylogenomic tree, SpecI results and ANI/AAI values we thus propose three new species of *Endozoicomonadaceae*: Candidatus Endozoicomonas pocilloporae sp. nov., *Candidatus Parendozoicomonas poriteae* sp. nov and *Candidatus Endozoicomonas milleporae* sp. nov. Additionally, we propose three lineages within *E. pocilloporae*, Poc-SVD1, Poc-SVD3 and Poc-SVD5, named according to the host genetic lineages (Fig. 5c, Supplementary Fig. 4). We also propose 6 lineages within *P. poriteae*, also named according to lineages, of which Por-K3c2 was not host lineage specific (Fig. 5c).

We constructed a phylogenetic tree based on 16S rRNA gene sequences (ASVs) obtained from *Tara* Pacific amplicon sequencing data and references from the literature. The 16S rRNA gene tree had a similar topology than the marker gene tree (Fig. 5b) separating

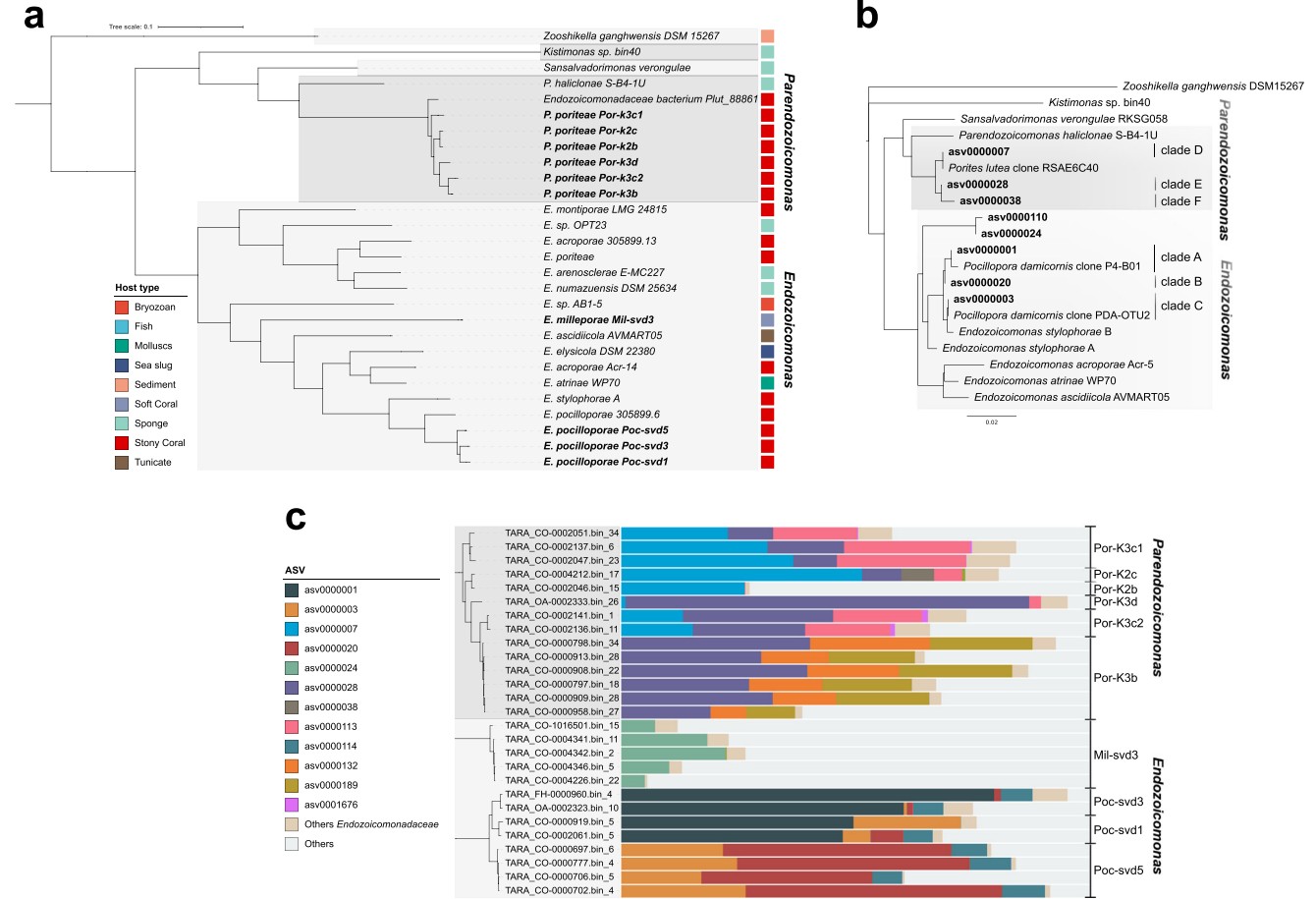

**Fig. 5 | Distance tree of Endozoicomonadaceae. a** Based on 71 concatenated single-copy marker proteins from MAGs and **b** based on 16S rRNA genes from ASVs. Clades were defined in this study. ASVs and MAGs from this study are in bold. **c** Proportion of ASVs in the samples from which the MAGs were reconstructed.

the 4 named *Endozoicomonadaceae* genera. ASVs from the 3 coral species were separated in host-specific clades under the *Endozoicomonas* and *Parendozoicomonas* genera. Within host-specific clades, ASVs grouped in distinct subclades. For *Pocillopora*, we delineated 3 clades. Clade A included the most common ASV (asv0000001), as well as sequences from *Pocillopora* of the Red Sea (clone P4-B01, KC668593[36]). Clade A was found in most *Pocillopora* genetic lineages (Supplementary Fig. 5). Clade B contained asv0000020 found mostly in the genetic lineage SVD5. Clade C contained asv0000003, as well as sequences from the Great Barrier reef (clone PDA-OTU2, AY700600[30]) (Fig. 5b), and was found mostly in host lineages SVD1, SVD2 and SVD5 (Supplementary Fig. 5). For *Porites*, we delineated 3 groups. The most common ASV was in clade D (asv0000007) together with a sequence from the Indian Ocean (clone RSAE6C40, KF180129[61]); it was more common in genetic lineage K2. Other common ASVs were in clade E (asv0000028), clade F (asv0000038) that were more common in the *Porites* lineage K3 (Supplementary Fig. 5). *Millepora* ASVs grouped in two clades, clade G and H, corresponding to the two most common ASVs (asv0000024 and asv0000110) (Fig. 4b).

Since the *Endozoicomonadaceae* MAGs did not contain 16S rRNA genes, we could not directly link the genomes to the ASVs. However, by looking at the ASV relative abundance in the samples from which the MAGs were reconstructed, we could associate groups of ASVs to the MAGs (Fig. 5c). In *Pocillopora*, the *Endozoicomonas* strains Poc-SVD1 and Poc-SVD3 corresponded to the dominance of asv0000001 (clade A), and Poc-SVD5 contained a majority of asv0000020 (clade B), but also asv0000003 (clade C). In *Porites*, Por-K3b, Por-K3c2 and Por-K3d had a majority of asv0000028 (clade E), while Por-K2b, Por-K2c and Por-K3c had more asv0000007 (D) (Fig. 5c).

### Endozoicomonadaceae mOTUs and mTAGs

*Endozoicomonadaceae* MAGs were also grouped into marker genes-based operational taxonomic units (mOTUs). All *Parendozoicomonas* MAGs from *Porites* grouped within one mOTU (mOTU_3), *Endozoicomonas* MAGs from *Pocillopora* were separated in two mOTUs (mOTU_2 and mOTU_67) and *Millepora* MAGs in one mOTU (mOTU_23) (Supplementary Fig. 6). Only one *Endozoicomonas* MAGs belonged to the mOTU_67 and was recovered from a SVD5 lineage coral while five of them grouped into the mOTU_2 and were recovered from SVD1 and SVD3 lineage corals (Supplementary Fig. 5). Finally, two MAGs did not have enough marker genes (<6) to be grouped into mOTUs. The mOTUs were specific to the coral host.

The relative abundance of *Endozoicomonadaceae* were determined based on mOTUs and mTAGs, which are 16S rRNA gene sequences extracted from metagenomes, in our three different coral species from 11 islands across the Pacific Ocean (Supplementary Fig. 7). We observed broadly similar patterns between mOTUs, mTAGs and ASVs profiles. We detected, however, some small differences: mTAGs and ASVs allowed the detection of *Endozoicomonadaceae* at very low abundance that were not seen with mOTU in Coiba (I02) and Guam (I15) islands for *Pocillopora*, in Coiba (I02), Cook (I08) and Rapa Nui (I04) for *Porites* and in Moorea (I07), Cook (I08), Niue (I09) and Guam (I15) islands for *Millepora* (Fig. 1, Supplementary Fig. 7).

### Metabolic capabilities of *Endozoicomonadaceae*

Among the pathways common to all *Endozoicomonadaceae* we noted the Embden-Meyerhof pathway, the pyruvate dehydrogenase, the citrate cycle, the non-oxidative phase of the pentose phosphate pathway, and the Type I, II secretion systems. Sequences coding for

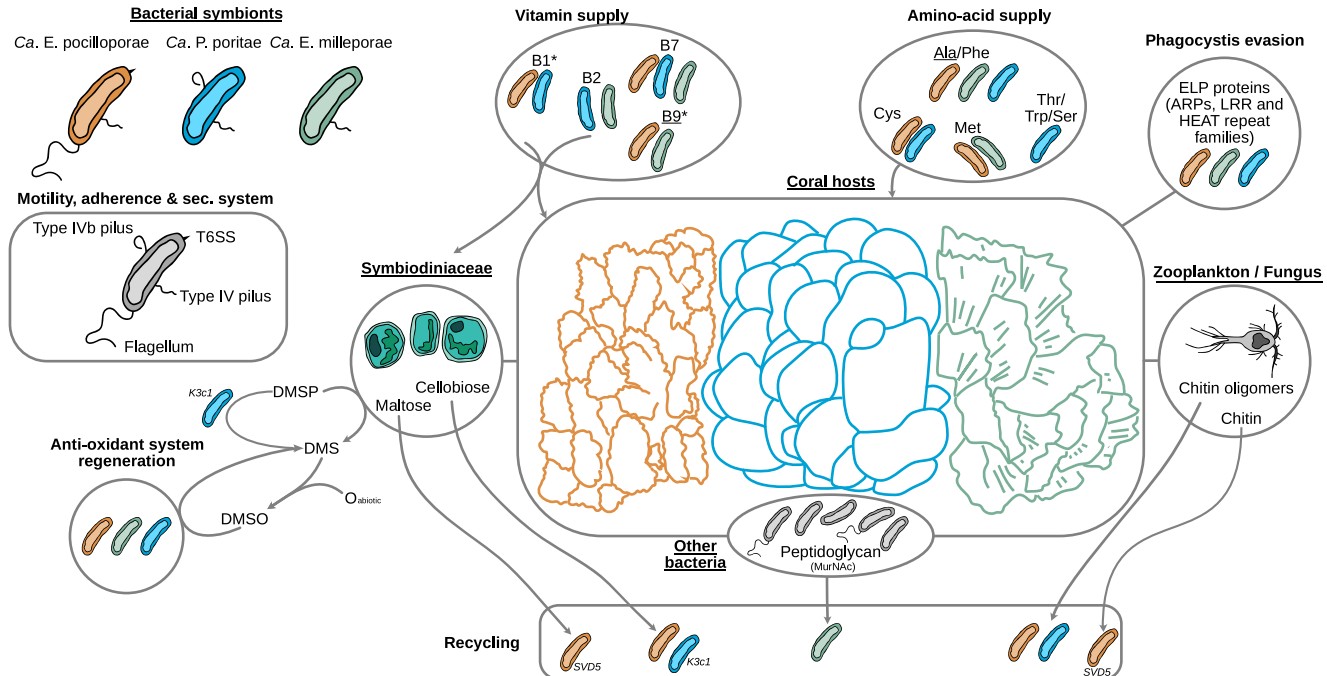

**Fig. 6 | Schematic summary of the inferred metabolisms and appendages characterizing the three new species and lineages (SVD5, K3c1) of *Endozoicomonadaceae* found in *Pocillopora* (orange), *Porites* (blue), and *Millepora* (green).** Underlined names: vitamin or amino acid also produced by the coral host, asterisk: genes for export not known. T6SS: type 6 secretion system, ELP: eukaryotic like protein, ARPs: actin-related proteins, LRR: leucine-rich repeats, DMSP: dimethyl-sulfoniopropionate, DMSO: dimethylsulfoxide, DMS: Dimethyl sulfide.

subunits of the DmsABC enzyme, which catalyzes the reduction of dimethylsulfoxide (DMSO) to dimethyl sulfide (DMS), were also detected in all *Endozoicomonadaceae* (Fig. 6, Supplementary Fig. 8). Most *Endozoicomonadaceae* encoded different eukaryotic-like proteins (ankyrin, leucine-rich, tetratricopeptide, HEAT and WD40 repeats). We did not find any genes involved in the assimilation of nitrate in the *E. pocilloporae, P. poriteae*, and *E. milleporae*, species, but they were present in *E. acroporae* and *P. haliclonae* (Supplementary Fig. 8).

The three new *Endozoicomonadaceae* genomes had genes for biotin (vitamin B7) biosynthesis, from pimeloyl-Coa and the pimeloyl-[acp], and export (Supplementary Fig. 9a). The biosynthesis pathway of the pimeloyl precursor through the BioI, BioW and BioC-BioH pathways were, however, incomplete in all the *Endozoicomonadaceae* genomes. The genes to synthesize this precursor through BioI (long chain acyl pathway) were present in all the genomes of the coral hosts that we analyzed (Supplementary Fig. 9b).

Most of the *Endozoicomonadaceae* genomes had genes for the biosynthesis of riboflavin (vitamin B2). *E. milleporae* and *P. poriteae* had the potential to export riboflavin, but not *E. pocilloporae* (Supplementary Fig. 9, Fig. 6). Genes for the synthesis of the two flavoproteins, derived from riboflavin, the flavin adenine dinucleotide (FAD) and the flavin mononucleotide (FMN) were found in all three *Endozoicomonadaceae* (Supplementary Fig. 9). The genes for riboflavin synthesis were not present in our coral host genomes (Supplementary Fig. 9b).

*E. pocilloporae* and *P. poriteae* also had the potential to synthesize thiamine (vitamin B1), while the coral host could not. Finally, *E. pocilloporae* and *E. milleporae* and the coral hosts had the potential of folate (vitamine B9) synthesis (Supplementary Fig. 9b).

Alanine and phenylalanine biosynthesis, and their potential export by genes with an AlaE and EamA domain, were found in the three *Endozoicomonadaceae* species (Fig. 6, Supplementary Fig. 10). Only *E. pocilloporae* and *P. poriteae* harbored genes for the cysteine synthesis, and had the potential to export it through the cydDC complex or bcr gene. Only *E. pocilloporae* and *E. milleporae* had genes involved in the

biosynthesis of methionine as well as genes with AzlCD domains, which could allow the export of the amino acid. Genes for serine (Ser) biosynthesis were present in *P. poriteae* and *E. pocilloporae* Poc-SVD5 and Poc-SVD3. Threonine (Thr) biosynthesis were present in some *P. poriteae* and *E. pocilloporae*. However, genes for export with ThrE domains (Thr and Ser export) were only found in *P. poriteae*. Tryptophan (Trp) biosynthesis genes were found only in *P. poriteae* that also had the gene carrying the EamA domain for putative Trp export.

The coral hosts only had the potential for alanine synthesis (Supplementary Fig. 10b).

Regarding motility, the presence of genes coding for pili and flagella varied between *Endozoicomonadaceae*. *E. milleporae* and *P. poriteae* did not have sequences for the flagella genes, while E. pocilloporae did. *E. milleporae* and *P. poriteae* had genes for the type IVb pilus, but *E. pocilloporae* did not. All had genes for the type IV pilus (Fig. 6). Some *Endozoicomonadaceae*, like *E. montiporae* and *P. haliclonae*, had all the genes for these three appendages (Supplementary Fig. 8). Type VI secretion system (T6SS) was only found in *E. pocilloporae* (Fig. 6).

Among genes coding for transporters, phosphate, zinc and methionine transporters were found in the *Endozoicomonadaceae* genomes from the 3 coral species (Supplementary Fig. 8). Inversely, sequence coding for the transport of Arginine were found only in *P. poriteae*, and L-Amino acid transport genes only in *E. pocilloporae* (Supplementary Fig. 8).

Genes for different subunits of the phosphotransferase system (PTS) were present in *Endozoicomonadaceae*. Most of the *Endozoicomonadaceae* species had the potential to assimilate fructose, glucose and beta-glucoside. Alpha-glucoside, trehalose and maltose PTS genes were found in *E. pocilloporae* and *P. poriteae* (Supplementary Fig. 8). *E. pocilloporae* and the Por-K3c1 clade of *P. poriteae* could assimilate cellobiose, while maltose assimilation potential was only present in the Poc-SVD5 clade within *E. pocilloporae*.

Protein coding sequences for the degradation of chitin residue were detected in *E. milleporae*, *E. pocilloporae* and *P. poriteae* species.

However, among chitin residues, *E. milleporae* has only the potential to assimilate a N-Acetylglucosamine (GlcNac) monomer, while *E. pocilloporae* and *P. poriteae* have the capabilities to assimilate GlcNac dimers (chitobiose) and trimers (chitotriose) (Supplementary Fig. 8). Although *E. pocillporae* and *P. poriteae* have the capability to actively import chitin oligomers (trimmer to hexamer) into their periplasm, only *E. pocilloporae* Poc-SVD5 has the chitinase gene to degrade GlcNac tretramer (Chitodextrine) and larger polymers. Among amino sugar metabolisms, N-Acetylmuramic acid (MurNac) phosphotransferase system was only detected in *E. milleporae* MAGs. MurNAc and GlcNAc can be both used for the biosynthesis of peptidoglycan, however, GlcNAc contrary to MurNAc can be redirected to the glycolysis pathway.

Genes involved in the uptake of the L-Ascorbate (vitamin C) and its anaerobic degradation in D-Xylulose-5P were found in *E. milleporae* and *P. poriteae* only (Supplementary Fig. 8). The type II L-ascorbate degradation (aerobic) was partially complete. In fact only the genes common to anaerobic degradation were present. Key genes required for degrading testosterone were not detected in the *E. milleporae*, *E. pocilloporae,* and *P. poriteae* species. The Endozoicomonadaceae *E. montiporae* and *E. elysicola* had all genes required for the degradation of androstenedione, but *E. pocilloporae*, *E. milleporae*, and *E. milleporae* did not (Supplementary Fig. 8).

*E. milleporae*, *P. poriteae*, and *E. pocilloporae* like *E. elysicola* genomes showed the potential for arsenate detoxification through the Type III arsenate detoxification and the arsenite methylation pathways (Supplementary Fig. 8).

Almost all Endozoicomonadaceae species had the capabilities of heme B biosynthesis from the L-glutamate (Supplementary Fig. 8). However, for *E. milleporae*, the first protein involved in the pathway, which catalyzes the reaction of transforming L-Glutamate in L-Glutamate-tRNA was not found. Additionally, none of our recovered MAGs contains the genes allowing siroheme biosynthesis. The genes implied in heme A biosynthesis from the heme were found in all Endozoicomonadaceae species except for *E. milleporae*. The biosynthesis of biliverdin, another product of the heme, was found in most of the *Endozoicomonadaceae* species apart from *E. milleporae*, Kistimonas and genomes from the unnamed *Endozoicomonadaceae* genus (Supplementary Fig. 8).

We also annotated the MAGs against the CAZy database specifically targeting carbohydrate active enzymes. A PCA based on CAZy composition separated the *Porites* MAGs from *Millepora* and *Pocillopora*, which could not be separated from each other (Supplementary Fig. 11). The CAZy that best explained the separation between groups are listed in Supplementary Data 2.

## Discussion

The unprecedented systematic sampling effort of the *Tara* Pacific expedition revealed that each studied coral genus was closely associated with one specific *Endozoicomonadaceae* species with broad cross-ocean geographic distribution despite varying environmental conditions. These new host-specific bacteria were composed of different lineages that varied in proportion between reefs, and the lineages' biogeography showed that the host-bacteria association patterns differed between host species. *Endozoicomonadaceae* strains had specific metabolic potentials, and they were associated with the host's genetic lineage, or the islands, rather than to environmental conditions.

### Pocillopora

*Pocillopora* was the coral genus that contained the highest proportion of *Endozoicomonadaceae*, and more precisely, it was specifically associated with the newly described species *E. pocilloporae* within the genus *Endozoicomonas* (Supplementary discussion). Among the three main *E. pocilloporae* strains, the most common, clade A (asv0000001, 90% of

the *Pocillopora* samples), appears like an extremely successful coral-associated bacterium. It was abundant across the entire Pacific Ocean, and also found, although at very low relative abundance, in *Porites* and *Millepora* (in 82% of all coral samples), suggesting an important function in the coral holobiont (explored further below). The clade A sequence was similar to sequences found earlier in *P. verrucosa* and *P. damicornis* in the Red Sea and American Samoa[36,43] indicating the presence of a global *E. pocilloporae* strain shared between different *Pocillopora* species. We also identified sequences 100% identical to our clade A in earlier studies on the effect of stress on coral microbiomes. Our clade A corresponded to an abundant *Pocillopora* OTU that did not decrease in abundance under elevated nitrate and urea concentrations in the Pacific Ocean (Moorea) or the Red Sea[42,55], although it decreased under excess dissolved organic carbon (OTU2[55]) and elevated temperature (OTU-Endo[62]). The success of *E. pocilloporae* clade A may thus, in part, rely on its tolerance to specific environmental perturbations, but it could also benefit from its ability to disperse and effectively colonize corals. We found it abundant in the plankton surrounding the coral colonies, or 'coral ecosphere'[22], and as far away as in water off the islands, yet in lower abundances. In addition, the genomes of *E. pocilloporae* harbour flagellar assembly genes, which are not found in *P. poriteae* and *E. milleporae*. The potential ability to produce a flagellum could allow bacteria to swim towards hosts[63,64], and facilitate effective colonization of corals able to spawn like *Pocillopora*, which can acquire prokaryotes from the environment[56,65–68]. Finally, *E. pocilloporae* was also characterized by the presence of a type VI secretion system, an antibacterial apparatus delivering toxins[69], that would give a competitive benefit by preventing other bacteria from multiplying in the coral host[70].

Clade A was, however, not always dominant. It was thought that *Pocillopora* was always associated with a single *Endozoicomonas*[43], but our study reveals that other clades (16S rRNA gene clade B and C) replaced the dominant clade A in some islands. These clades were associated with specific genetic lineages of *Pocillopora*. Clade B (asv0000020) was most abundant in the genetic lineage SVD5 found in Easter Island (I04), where Clade A was completely absent. Clade B could represent a bacterium that has co-evolved with its SVD5 host. SVD5 was also present in Moorea (I07), but there, clade B was present in association with Clade A. The host-bacteria association could thus reflect an interplay between co-evolution that is strongly marked in isolated islands, such as Easter Island, and other selection factors leading to mixed communities, potentially via 'shuffling' of abundant *Endozoicomonas* phylotypes[71] in more interconnected islands. Clade C (asv0000003) was most abundant at the Great Barrier Reef (I19), and was closely related to a bacterium earlier detected in *P. damicornis* in eastern Australia[30]. Our field observations indicate that the corals sampled at the Great Barrier Reef during the *Tara* Pacific expedition may have been *P. damicornis*. Clade C could thus be specific to that coral species. Clade B and C had, together with clade A, the potential for type IV pili synthesis, which could help host sensing and surface motility[72].

Our genome analysis showed that *E. pocilloporae* have genes coding for the production of the essential vitamins B1, B2 and B7, and specific amino acids (Phe, Cys, Met), while the coral hosts did not have these metabolic capabilities. The coral holobiont must thus acquire these essential metabolites from heterotrophic feeding and/or its associated prokaryotes, including *E. pocilloporae*. Many algae like Symbiodiniaceae are auxotrophs for different B vitamins[12,56,73,74] and could also need vitamins produced by bacteria.

Our data also provide clues about *E. pocilloporae* lifestyle. They had the genomic potential for cellobiose assimilation, suggesting that they could use cellulose originating from the degradation of Symbiodiniaceae cells[23]. However, our study also revealed differences in potential metabolisms between *E. pocilloporae* lineages. Poc-SVD1, contrary to Poc-SVD5 and Poc-SVD3, had the potential to degrade chitin, the most abundant polysaccharide in the ocean[75], and which for instance could originate from zooplankton taken by the coral via

heterotrophic feeding. The Poc-SVD5 genomes, corresponding to 16S rRNA gene Clade B and C defined in this study, had the potential to assimilate maltose, a sugar resulting from the hydrolysis of starch. Starch could originate from Symbiodiniaceae's carbon and energy storage. Differences in maltose metabolisms between different *E. pocilloporae* lineages may thus reflect the presence of different Symbiodiniaceae genera that have different energy storage strategies, and thus different starch content[76]. Poc-SVD5 was associated with Symbiodiniaceae Cladocopium C42, while Poc-SVD1 was reconstructed from samples with higher proportion of Cladocopium C1.

### Porites

In *Porites*, although *Endozoicomonas* was the most common and abundant bacterial symbiont, it was rare or absent from a large number of islands. *Endozoicomonas* has earlier been shown to be abundant in *Porites* from the Caribbean[34,38] and the Pacific Ocean[53], but our data reveal that their overall abundance can vary greatly geographically. The pattern of host-bacteria association thus appears different from the one seen in *Pocillopora*, with different species of associated *Endozoicomonadaceae*. The new species *P. poriteae*, genus *Parendozoicomonas*, was dominated by Clade D (asv0000007) found from the Gambier Island (I06) to Sesoko Island in the west Pacific (I19). Clade D sequences were similar to the ones found earlier in the Indian Ocean off South Africa[61] indicating that its geographic spread is probably large. This wide geographic distribution, although not as extensive as *E. pocilloporae* Clade A, may also be due to resistance to some environmental perturbations. Sequence comparison showed that *Endozoicomonas* similar to clade D resisted elevated environmental nitrate and urea concentrations, and elevated temperatures[42]. In *P. lobata*, *Endozoicomonadaceae*, although we could not verify lineage identity, did not decline under nutrient stress[77] or elevated pH at $CO_2$ seeps[48,53]. Other *P. poriteae* lineages were more limited in their distribution, and contrary to *E. pocilloporae*, the distribution of these less abundant lineages was associated with islands rather than their host's genetic lineage. Clade F was for instance mostly restricted to Ducie Island (I05) and Cook Island (I09).

The *P. poriteae* genomes also showed differences to *E. pocilloporae*; they had genes coding for a type IVb pilus for host colonization and adhesion[78], and may be able to provide their host with B1, B2 and B7 vitamins, and Thr/Trp amino acids. Interestingly, there were also differences among the different *P. poriteae* lineages. Por-K3c1, which corresponded to asv0000007, was the only *P. poriteae* that had the potential for cellobiose assimilation. The Por-K3c1 lineage was also the only one to have a gene coding for metabolizing dimethylsulfoniopropionate (DMSP) into dimethyl sulfide (DMS). That gene was not found in other *P. poriteae*, *E. pocilloporae* or *E. milleporae* lineages, which indicates that the involvement of *Endozoicomonadaceae* in the sulfur cycle[24] is very much lineage specific.

### Millepora

In *Millepora*, host-bacteria association differed from that of scleractinian corals, with only few sampled fire coral colonies having abundant communities of *Endozoicomonas*. The association seems looser and thus probably less important for the host. Among the 2 main *E. milleporae* lineages, clade G had the widest distribution, while clade H was more common in areas around New Caledonia (I21). Our analyses suggest that *E. milleporae* clades were associated with specific Symbiodiniaceae genera and coral host genetic lineages, but these results remain hypothetical since only few *Endozoicomonas* were present in the 11 islands for which host genetic information was available. Our results contrast with the ones from a recent study in Moorea[47] where differences in bacterial abundance were associated with reef habitats rather than host genotype. Similarly, in the soft coral *Scleronephthya gracillium*, *Endozoicomonas* abundance varied greatly along the Kuroshio Current Region, and was seemingly related to differences in reef

habitats[79]. In our study, the environmental parameters did not strongly explain the variability of *Endozoicomonas* in *Millepora*. Regarding potential metabolisms, *E. milleporae*, contrary to *E. pocilloporae and P. poriteae*, did not have the potential to use GlcNac polymer, cellobiose or starch, but could use peptidoglycans from other bacteria.

Our results draw a fine picture of the patterns of *Endozoicomonadaceae* abundance in corals across the Pacific Ocean with implications for our understanding of the ecology and evolution of host-symbiont relationships. We show that the ecology of this widespread symbiont should be considered at the lineage level to understand the factors structuring communities and infer associated metabolic contributions. Our data suggest that different coral species exhibit distinct host-*Endozoicomonadaceae* relationships ranging from a strong association illustrated by the global, and abundant presence of *Endozoicomonas* in *Pocillopora*, to a rather weak association with a rare and scattered presence in *Millepora*. In all corals, the environment had generally only a small structuring effect on *Endozoicomonadaceae* community composition, while the genetic lineage of the host was important in some corals, arguing for a high level of host specificity putatively shaped by long co-evolutionary histories. Thus, coral-bacterial association at large may range from stable co-dependent relationships that arose through evolutionary time to opportunistic associations that are flexible and determined by the prevailing environment. Even for bacteria within the family *Endozoicomonadaceae*, we find evidence for one and the other. *Pocillopora* was associated with a very abundant *Endozoicomonas* lineage throughout its entire geographic range. The success of this lineage could be explained by its functional potential inferred from genome characteristics including the ability to swim, destroy other bacteria, and thrive on carbon sources produced by Symbiodiniaceae. Other *E. pocilloporae* lineages were specific to the host's genetic lineages, while in *P. poriteae*, the lineages were linked to the location (island). This unique dataset from the *Tara* Pacific expedition reveals the taxonomic and functional diversity of coral-associated *Endozoicomonadaceae* and emphasizes the need for studies at the lineage level to uncover the bacterial compartment of the coral holobiont.

## Methods

### Sample collection, environmental parameters, and biomarkers

Coral and plankton samples were taken across the Pacific Ocean from three to four different sites at each of the 32 islands visited during the *Tara* Pacific expedition[80]. The detailed sampling protocols are presented in Lombard et al.[81]. Based on morphology, we targeted the complex *Pocillopora meandrina*, the robust *Porites lobata* and the fire coral *Millepora platyphylla*. Samples from 10 different colonies from each species were collected at each site using a hammer and a chisel. They were stored individually underwater in Ziploc bags, and conditioned in tubes with DNA/RNA shield (Zymo Research, Irvine, CA, USA) at −20 °C once on board. Additionally, plankton was sampled on 0.2–3 µm and 3–20 µm filters by filtering 100 L of water obtained with a pump near the islands and over the reefs, and with a diver held hose at the surface of the *Pocillopora* colonies. All filters were preserved in cryovials in liquid nitrogen. Because some corals within the same genus can be difficult to differentiate by eye, results are aggregated at genus level: *Pocillopora*, *Porites* and *Millepora*. Several environmental parameters were measured at the time of collection following the protocols detailed in Lombard et al.[81]. These include temperature and salinity measured with a CTD (Castaway CTD), nutrients quantified back in the laboratory and chlorophyll a (chla) concentrations. Sample provenance and environmental context are available on *Zenodo*[82].

### DNA extraction and sequencing

DNA was extracted with comercial kits after mecanical cell disruption. The detailed protocols for corals and plankton are presented in Belser et al.[83]. Metabarcoding sequencing targeted all 10 colonies taken at

each site for each coral genus at the 32 islands. The prokaryote 16S rRNA gene was first amplified with a nested PCR approach with a first full-length amplification using the 27F/1492R 16S rRNA prokaryotic primer set[29,84] in order to increase the target prokaryotic DNA, and a second amplification using the 515F-Y/926R primers[85]. The Symbiodiniaceae internal transcribed spacer (ITS2) was amplified with primers SYM-VAR-5.8S2 / SYM-VAR-REV. The sequencing was performed on an Illumina sequencer to obtain 0.5 to 1 M of ~250 pb 16S rRNA paired-end reads[83], and ~20.000 of ~250 bp paired-end reads for ITS2. Details are provided in Belser et al.[83].

Metagenomic sequencing was performed on three coral replicates from each site of a subset of 11 islands (I01-Islas de las Perlas, I02-Coïba, I03-Malpelo, I04-Rapa Nui, I05-Ducie, I06-Gambier, I07-Moorea, I08-Aitutaki, I09-Niue, I10-Upolu, and I15-Guam). The sequencing was performed on a NovaSeq6000 or HiSeq4000 Illumina sequencer (Illumina, San Diego, CA, USA) in order to produce 100 M of ~150 pb paired-end reads per sample. Details are provided in Belser et al.[83].

All sequencing files were submitted to the European Nucleotide Archive (ENA) at the EMBL European Bioinformatics Institute (EMBL-EBI) under the *Tara* Pacific Umbrella project PRJEB47249. Samples and their metadata were registered in the ENA biosample database. Sample provenance and environmental context are available on *Zenodo*[82].

## 16S rRNA gene and ITS2 sequence analysis
For 16S rRNA gene analysis[15], adapters were removed with cutadapt (version 2.8)[86] and an ASV abundance table was built with DADA2 (version 1.14)[87] as detailed in the scripts published in *Zenodo*[88]. Samples were grouped by sequencing lane to learn errors and infer ASVs. ASVs representing less than six inserts were tagged as being spurious and removed. Taxonomic annotation was performed with IDTAXA[89] with a confidence threshold of 40 against the SILVA v.138 database. Eukaryotic ASVs (chloroplast and mitochondria) were identified based on taxonomic annotation following the criteria published in *Zenodo*[88] and removed from the dataset prior to analysis. In addition, bacterial sequences annotated at the family level as Oxalobacteraceae, Comamonadaceae, Cutibacterium and Yersiniaceae were identified as reagent contaminants and removed from the dataset.

Symbiodiniaceae profiles were obtained using SymPortal[90], which uses the intragenomic diversity of Symbiodiniaceae ITS2 to define ITS2 type profiles based on consistent co-occurrence of intragenomic ITS2 variants across all samples. A hierarchical clustering, using *hclust* function from the R "stats" package with Complete Linkage method, was performed on the unifrac matrix distance of the ITS2 abundance table to obtain Symbiodiniaceae community clusters.

## Host lineage assignation and phenotypic signature
A set of genome-wide unlinked single nucleotide polymorphisms (SNPs), represented in >100 colonies of *Pocillopora* and *Porites*, were identified in the metagenomics dataset from the subset of 11 islands. For *Millepora* only target genes SNP of around 60 colonies were produced. From the analysis of these SNPs[91] we identified five independent genetic lineages in *Pocillopora*, three in *Porites* (nine geographic sub-lineages) and five in *Millepora*. The detailed method for host lineage assignment protocols are presented in ref. 91.

The phenotypic signatures of *Pocillopora* and *Porites* were defined through six biomarkers: the coral protein content, the Symbiodiniaceae content per coral mg of protein biomass[92], Symbiodiniaceae content per coral surface[92], protein ubiquitination[93] and carbonylation per coral mg of protein[94] to assess the cellular damages, and total antioxidant content per coral mg of protein to assess the cellular defenses[95]. The detailed protocols are presented in ref. 96.

## 16S rRNA gene data analysis
A total of 2447 metabarcoding samples of corals were analyzed. Prokaryotic and *Endozoicomonadaceae* community diversity were estimated by computing the Shannon index with the function *diverse* from "diverse" R package[97]. In order to account for the compositional characteristic of the prokaryote abundance table, data were weighted center log ratio transformed (wCLR) before computing statistics. To infer the effect of environmental factors and coral phenotypic biomarkers on *Endozoicomonadaceae* community composition, Kendall's rank correlation was computed for the 32 islands using *bioenv* function from "vegan" R package[98] and *p*-value were adjusted using Benjamini et Hochberg[99] method.

The correlations between *Endozoicomonadaceae* community composition and environmental factors, phenotypic biomarkers, coral genetic lineage, or Symbiodiniaceae assemblage were tested individually with a mantel test[100] using *mantel* function from "vegan" R package. The matrices were based on Euclidean distance for all datasets except for the genetic lineages that were based on RaxML[101] distances. They were computed for *Pocillopora* and *Porites* on the subset of 11 islands for which data on genetic lineages and phenotypic biomarkers were available.

A variance partitioning analysis was done, using the function *fitExtractVarPartModel* from the "variancePartition" R package[102], to quantify the contribution of the location (island), the coral genetic lineage, and the Symbiodiniaceae assemblage (community clusters) on *Endozoicomonadaceae* community composition across the subset of 11 islands.

An ASVs distance tree was constructed using 16S rRNA gene sequences. Sequences were aligned with MUSCLE[103], and the alignment was trimmed manually to remove gaps and get all sequences to start and end at the same position. A distance matrix was computed with the KIMURA model of transition and transversion rates, and the tree constructed based on the Fitch-Margoliash method, based on least squares principle, in PHYLIP[104].

## Metagenomic assemblies and binning
A total of 270 metagenomes (101 *Pocillopora*, 108 *Porites* and 61 *Millepora*) were analyzed. Sequencing reads were quality filtered using BBMap (version 38.71)[105] by removing sequencing adapters from the reads, removing reads that mapped to quality control sequences (PhiX genome) and discarding low-quality reads using the parameters trimq = 14, maq = 20, maxns = 0 and minlength = 45. Downstream analyses were performed on quality-controlled reads, or, if specified, merged quality-controlled reads (bbmerge.sh minoverlap = 16). Quality-controlled reads were normalized (bbnorm.sh target=40, mindepth = 0) before they were assembled with metaSPAdes (version 3.12 or version 3.13 if required)[106]. The resulting scaffolded contigs (hereafter scaffolds) were finally filtered by length (≥1 kbp). The quality-controlled metagenomic reads from all samples were individually mapped against the scaffolds of each sample (270 * 270 mapping) using the Burrows-Wheeler-Algorithm (BWA)(version 0.7.17-r1188)[107]. Alignments were filtered to be at least 45 bases in length, with an identity of ≥97% and covering ≥80% of the read sequence. The resulting BAM files were processed using the jgi_summarize_bam_contig_depths script of MetaBAT2 (version 2.152.1)[108] to provide within- and between-sample coverages for each scaffold.v5. The scaffolds were finally clustered into bins for each metagenomic sample with the automatic algorithm MetaBAT2 (version 2.15) using parameters −minContig −2000 and maxEdges 500 to increase sensitivity. The quality of each metagenomic bin was evaluated using both the 'lineage workflow' of CheckM (version 1.1.1)[109] and Anvi'o (version 6.1)[110]. Metagenomic bins for which either CheckM/Anvi'o reported a completeness/completion ≥30% and a contamination/redundancy ≤10% were taxonomically annotated using GTDB-Tk (version 1.0.2)[111] with default parameters against the GTDB r89 release[112] in order to select *Endozoicomonadaceae* bacteria. The 27 identified bins are available on Zenodo (https://zenodo.org/record/7840163).

### Species-level profiling with mOTUs and mTAGs

Metagenomic bins that passed the quality filtering were additionally added to the database (version 2.5.1) of the metagenomic profiling tool mOTUs[113] to generate an extended mOTUs reference database, following the approach of Paoli et al.[114]. Only genomes with at least six out of the ten universal single-copy marker genes as identified by fetchMGs (version 1.1)[115] were used. The species-level profiling was then computed by mapping quality-controlled metagenomic reads against this extended database using default parameters of mOTUs (version 2).

Taxonomic profiling using degenerate consensus reference sequences of ribosomal RNA genes was performed with mTAGs (version 1.0)[116] on metagenomic reads samples.

### Selection of *Endozoicomonadaceae* from databases

All available *Endozoicomonadaceae* genomes (24) and closely related (two *Zooshikella*) were downloaded from the NCBI genomes database (last accessed November 2021). Additionally, five *Endozoicomonas* genome assemblies from Neave et al. 2017[23] were recovered from the authors and two putative *Endozoicomonadaceae* genomes (CAR1_bin20 and CAR3_bin23) assembled in Robbins et al.[117] reobtained from https://data.ace.uq.edu.au/public/sponge_mags. External genomes were taxonomically checked using GTDB-Tk as described for the metagenomic bins.

### Quality evaluations of *Endozoicomonadaceae* bins and genomes

The quality of each *Endozoicomonadaceae* bins and added genomes from the literature was assessed using the 'lineage workflow' of CheckM (version 1.1.1)[109] and Anvi'o (version 6.1)[110]. *Endozoicomonadaceae* bins with a completeness ≥50% and a contamination ≤10% were kept for downstream analysis and will be referred to as Metagenome Assembled Genomes (MAGs) or genomes.

The three new *Endozoicomonadaceae* species described in the present study (*Endozoicomonas pocilloporae*, *Parendozoicomonas poriteae* and *Endozoicomonas milleporae*) have been deposited in the SeQCode registry[118].

### Genes calling and annotation

Gene calling was performed using Prodigal (version 2.6.3)[119] and ribosomal RNAs were extracted using HMMER (version 3.3.2)[120]. These steps were performed with Anvi'o (version 6.1) by following the procedure described here: https://merenlab.org/2016/06/22/anvio-tutorial-v2. The predicted proteins recovered from *Endozoicomonadaceae* genomes were annotated by identifying protein domain family with HMMER searches (http://hmmer.org/) against the PfamA database (version 33.1)[121] and by assigning orthologous groups (OGs) using eggNOG-mapper (version 2.0.1)[122,123] based on eggNOG database (version 5.0)[124]. KEGG[125] and InterProscan[126] annotation was inferred from the OGs annotation. Based on this annotation, completion of metabolic modules/pathways was accessed using the information available from KEGG Metabolic pathways[125] and MetaCyc Metabolic Pathway Database[127]. Annotation of the predicted proteins was also performed using the HMMs from dbCAN (Version 11) to identify Carbohydrate-Active Enzymes (CAZy)[128].

### Comparative genomic analysis

The phylogenomic tree of *Endozoicomonadaceae* was inferred using a concatenated alignment of 71 bacterial single-copy marker proteins as detailed by the Anvi'o (https://anvio.org/). The tree included MAGs from this study as well as reference genomes from earlier studies that had ≥61 single-copy markers. The tree was visualized and annotated using the iTOL (version 6.4.1)[129]. Average nucleotide identity (ANI) between *Endozoicomonadaceae* genomes were calculated using PyANI implemented in Anvi'o (version 6.1). Pairwise amino acid identity (AAI) between genomes was computed using *comparem* tool (*similarity* and *aai* command) (https://github.com/dparks1134/CompareM). In order to provide a species-level clustering of genomes, dRep (version 3.1.1)[130] with a 95% ANI threshold[131] and SpecI[132] reused on all genomes. *Endozoicomonadaceae* MAGs within the same genome cluster were grouped into subclades based on mean ANI value ≥98% and host species lineage isolation source. Newly identified *Endozoicomonadaceae* species were named according to the host genus and subclade according to the genetic lineage of the host.

### Coral genomes

Coral genomes assembly and annotation protocols are detailed in Noel et al.[133]. Briefly, *Pocillopora* cf. *effusa*, *Porites lobata* colonies were collected at Moorea, French Polynesia. An amount of 15 g of each coral colony was stored at −80 °C, after flash freezing using liquid nitrogen, until DNA extraction. The genomes were sequenced using a combination of long (Oxford Nanopore Technologies, ONT) and short reads (Illumina). To generate long-reads based genome assemblies four different assemblers were used: Smartdenovo[134], Redbean[135], Flye[136], and Ra[137]. From the assemblies genes calling was performed and were functionally assigned by aligning them against the nr database[138] (NCBI) using diamond[139] (version 0.9.24).

### Reporting summary

Further information on research design is available in the Nature Portfolio Reporting Summary linked to this article.

## Data availability

Sample provenance and environmental context are available on *Zenodo*[82]. Endozoicomonadaceae MAGs are available on *Zenodo* (https://zenodo.org/record/7840163). Samples and their metadata were registered in the ENA biosample database. All sequencing files were submitted to the European Nucleotide Archive (ENA) at the EMBL European Bioinformatics Institute (EMBL-EBI) under the Tara Pacific Umbrella BioProject PRJEB47249. All other data supporting the findings of this study are provided in the Supplementary Information or the Source Data. Source data are provided with this paper.

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

## Acknowledgements

Special thanks to the *Tara* Ocean Foundation, the R/V *Tara* crew, and the *Tara* Pacific Expedition Participants (https://doi.org/10.5281/zenodo.3777760). We are keen to thank the commitment of the following institutions for their financial and scientific support that made this unique 3-year *Tara* Pacific Expedition possible: CNRS, PSL, CSM, EPHE, Genoscope, CEA, Inserm, Université Côte d'Azur, ANR, agnès b., the Veolia Foundation, the Prince Albert II de Monaco Foundation, Région Bretagne, Lorient Agglomération, L'Oréal, Biotherm, Billerudkorsnas, AmerisourceBergen Company, Oceans by Disney, France Collectivités, Fonds Français pour l'Environnement Mondial (FFEM), UNESCO-IOC, Etienne Bourgois, and the Tara Ocean Foundation teams. P.E.G. acknowledges the support of Agence Nationale de la Recherche (ANR) through the CORALMATES project (ANR-18-CE02-0009), and S.S. acknowledges the support of the Swiss National Science Foundation grant 205321_184955. C.H. is grateful to the genotoul bioinformatics platform, Toulouse, Occitanie (Bioinfo Genotoul) for providing computing resources (https://doi.org/10.15454/1.5572369328961167E12). *Tara* Pacific would not exist without the continuous support of the participating institutes. The authors also particularly thank Serge Planes, Denis Allemand, and the *Tara* Pacific consortium. This is publication number #24 of the *Tara* Pacific Consortium. Sampling permits are provided as supplementary material. CdV, FL and SP, are funded by the European Union's Horizon 2020 research and innovation programme "Atlantic Ecosystems Assessment, Forecasting and Sustainability" (AtlantECO) under grant agreement No 862923. This study was supported in part by FRANCE GENOMIQUE (ANR-10-INBS-09).

## Author contributions

C.H. and P.E.G. designed the experiments, analyzed the data, and wrote the manuscript; L.P., H.J.R., and G.S. conducted the bioinformatic analyses of the raw sequence data and constructed MAGs; M.Z., C.P., C.R.V., D.A.P.G., E. Boissin, L.P. provided useful comments and manuscript edition. C.H., L.P., H.J.R., G.S., E. Boissin, S. Romac, J.P., G.B., G.I., C.M., M.Z., B.P., E.J.A., B.C.C.H., J.M.A., C.P., D.A.P.G., M.M.N., S.A., B.B., E. Boss, C.B., C.d.V., E.D., M.F., D.F., P.F., E.G., F.L., S. Pesant, S. Reynaud, O.P.T., R.T., P.W., D.Z., D.A., S. Planes, R.V.T., C.R.V., S.S., and P.E.G. reviewed the manuscript.

## Competing interests

The authors declare no competing interests.

## Additional information

[1]Sorbonne Université, CNRS, Laboratoire d'Ecogéochimie des Environnements Benthiques (LECOB), Observatoire Océanologique de Banyuls, 66650 Banyuls sur Mer, France. [2]Department of Biology, Institute of Microbiology and Swiss Institute of Bioinformatics, ETH Zürich, 8093 Zürich, Switzerland. [3]PSL Research University: EPHE-UPVD-CNRS, USR 3278 CRIOBE, Laboratoire d'Excellence CORAIL, Université de Perpignan, 52 Avenue Paul Alduy, 66860 Perpignan, Cedex, France. [4]Sorbonne Université, CNRS, Station Biologique de Roscoff, AD2M, UMR 7144, ECOMAP, Roscoff, France. [5]Génomique Métabolique, Genoscope, Institut François Jacob, CEA, CNRS, Univ Evry, Université Paris-Saclay, Evry, France. [6]Research Federation for the study of Global Ocean Systems Ecology and Evolution, FR2022/Tara GOSEE, 75000 Paris, France. [7]School of Marine Sciences, University of Maine, Orono, ME, USA. [8]Fondation Tara Océan, 8 rue de Prague, 75012 Paris, France. [9]Department of Animal Ecology & Systematics, Justus Liebig University Giessen, Heinrich-Buff-Ring 26–32 (IFZ), 35392 Giessen, Germany. [10]CNRS, INSERM, Institute for Research on Cancer and Aging (IRCAN), Université Côte d'Azur, Nice, France. [11]Laboratoire International Associé Université Côte d'Azur-Centre Scientifique de Monaco (LIA ROPSE), Monaco, Principality of Monaco. [12]Department of Biology, University of Konstanz, 78457 Konstanz, Germany. [13]Centro de Investigaciones Biológicas del Noroeste (CIBNOR), La Paz, Baja California Sur 23096, México. [14]Shimoda Marine Research Center, University of Tsukuba, 5-10-1, Shimoda, Shizuoka, Japan. [15]School of Marine Sciences, University of Maine, Orono, ME 04469, USA. [16]Institut de Biologie de l'Ecole Normale Supérieure (IBENS), Ecole normale supérieure, CNRS, INSERM, Université PSL, 75005 Paris, France. [17]Laboratoire des Sciences du Climat et de l'Environnement, LSCE/IPSL, CEA-CNRS-UVSQ, Université Paris-Saclay, 91191 Gif-sur-Yvette, France. [18]Weizmann Institute of Science, Department of Earth and Planetary Sciences, 76100 Rehovot, Israel. [19]Department of Medical Genetics, CHU Nice, Nice, France. [20]Sorbonne Université, Institut de la Mer de Villefranche sur mer, Laboratoire d'Océanographie de Villefranche, 06230 Villefranche-sur-Mer, France. [21]Institut Universitaire de France, 75231 Paris, France. [22]European Molecular Biology Laboratory, European Bioinformatics Institute, Wellcome Genome Campus, Hinxton, Cambridge CB10 1SD, UK. [23]Centre Scientifique de Monaco, Monaco, Principality of Monaco. [24]School of Biological and Chemical Sciences, Ryan Institute, University of Galway, Galway, Ireland. [25]Microbiology Department, Oregon State University, Corvallis, OR, USA. ✉e-mail: pierre.galand@obs-banyuls.fr

