## [Peer review file · Nature Communications]

REVIEWERS' COMMENTS

Please note, Reviewer #1 is the same reviewer #1 from when your manuscript was under consideration at Nature Microbiology. Reviewer #3 is a new reviewer.

Reviewer #1 (Remarks to the Author):

The paper, now for consideration for publication in Nature Communications, is improved from the original submitted to Nature Microbiology and most reviewer comments were addressed. The track changes document did not track the author changes. There are substantially more authors in the resubmission. There are still issues with italicisation - sometimes Symbiodiniaceae is italicized and other times not. Although the authors will register their sequences through SeqCode and perhaps get Candidatus status for the sequences, the fact remains that Endozoicomonas are cultivable, albeit that some publications note some difficulty in achieving this with different strains. However, the co-authors of this current publication did not attempt cultivation, so cannot comment on the difficulty or otherwise of this endeavour. The Candidatus status method for achieving naming, should be approached after cultivation approaches are unsuccessful. The sequence data are always available for inclusion in any data analyses by these and other authors.

Reviewer #3 (Remarks to the Author):

This is a comprehensive examination of Endozoicomonaceae associated with Pacific corals, and includes both marker gene and genome reconstructions from metagenomes. It's a valuable resource to the coral and marine animal symbiont community which is well-written. I found the previous reviewer comments to be comprehensively addressed. I found a few minor areas for corrections.

Ln 278: Precision of the completeness of the MAGS down to 2 decimal places seems unlikely; change to 51.2%, 93.3% and contamination levels 0 – 4.9%. Also see lines 294, 306/307 for similar two decimal place representation situations.

Ln 470: Remove the extra period.

Ln 501: It is not clear to me what 16S rRNA clade B and C are – please remind us that these are your own designations. The lettered designations should also be explained in the legend of Figure 5.

Lines 585-586: Include the volumes of water sampled.

Reviewer #1 (Remarks to the Author):

The paper, now for consideration for publication in Nature Communications, is improved from the original submitted to Nature Microbiology and most reviewer comments were addressed.

The track changes document did not track the author changes.

- *We are sorry to read that. We double checked and it looks like the file that was on the Nat Com website does show the changes.*

There are substantially more authors in the resubmission.

- *The change in authors was due to a change in publication guidelines within the Tara Pacific consortium. The added authors' names were previously referred to as "Tara Pacific Consortium Coordinators" and were listed at the end of the manuscript.*

There are still issues with italicisation - sometimes Symbiodiniaceae is italicized and other times not.

- *We removed all italicisation for Symbiodiniaceae.*

Although the authors will register their sequences through SeqCode and perhaps get Candidatus status for the sequences, the fact remains that *Endozoicomonas* are cultivable, albeit that some publications note some difficulty in achieving this with different strains. However, the co-authors of this current publication did not attempt cultivation, so cannot comment on the difficulty or otherwise of this endeavour. The Candidatus status method for achieving naming, should be approached after cultivation approaches are unsuccessful. The sequence data are always available for inclusion in any data analyses by these and other authors.

- *We understand the point of view of the reviewer, and we are confident that our use of the term Candidatus follows the most recent guidelines.*

Reviewer #3 (Remarks to the Author):

This is a comprehensive examination of Endozoicomonaceae associated with Pacific corals, and includes both marker gene and genome reconstructions from metagenomes. It's a valuable resource to the coral and marine animal symbiont community which is well-written. I found the previous reviewer comments to be comprehensively addressed. I found a few minor areas for corrections.

Ln 278: Precision of the completeness of the MAGS down to 2 decimal places seems unlikely; change to 51.2%, 93.3% and contamination levels 0 – 4.9%. Also see lines 294, 306/307 for similar two decimal place representation situations.

- *Done.*

Ln 470: Remove the extra period.

- *Done.*

Ln 501: It is not clear to me what 16S rRNA clade B and C are – please remind us that these are your own designations. The lettered designations should also be explained in the legend of Figure 5.

- *We now specify that the two clades were defined in the study and the origin of the clades is now stated in the legend of figure 5.*

Lines 585-586: Include the volumes of water sampled.

- *We added the water sampling volumes.*